# Status of Infectious Diseases in Free-Ranging European Brown Hares (*Lepus europaeus*) Found Dead between 2017 and 2020 in Schleswig-Holstein, Germany

**DOI:** 10.3390/pathogens12020239

**Published:** 2023-02-02

**Authors:** Marcus Faehndrich, Jana C. Klink, Marco Roller, Peter Wohlsein, Katharina Raue, Christina Strube, Ellen Prenger-Berninghoff, Christa Ewers, Lorenzo Capucci, Antonio Lavazza, Herbert Tomaso, Joseph G. Schnitzler, Ursula Siebert

**Affiliations:** 1Institute for Terrestrial and Aquatic Wildlife Research, University of Veterinary Medicine Hannover, Foundation, 30559 Hannover, Germany; 2Department of Pathology, University of Veterinary Medicine Hannover, Foundation, 30559 Hannover, Germany; 3Institute for Parasitology, Centre for Infection Medicine, University of Veterinary Medicine Hannover, Foundation, 30559 Hannover, Germany; 4Institute of Hygiene and Infectious Diseases of Animals, Justus Liebig University Giessen, 35390 Giessen, Germany; 5Istituto Zooprofilattico Sperimentale della Lombardia e dell’Emilia Romagna, Via Bianchi 7/9, 25124 Brescia, Italy; 6Institute of Bacterial Infections and Zoonoses, Friedrich-Loeffler-Institut—Federal Research Institute for Animal Health (FLI), Naumburger Strasse 96a, 07743 Jena, Germany

**Keywords:** *Lepus europaeus*, population health, infectious diseases, EBHSV, *Yersinia pseudotuberculosis*, RHDV2, hepatitis, steatitis, coccidiosis, *Trichostrongylus* spp.

## Abstract

The European brown hare (*Lepus europaeus*) is a quite adaptable species, but populations have been decreasing for several decades in different countries, including Germany. To investigate infectious diseases as possible influences on observed population decline in the German federal state Schleswig-Holstein, 118 deceased free-ranging European brown hares were collected between 2017 and 2020 and underwent detailed postmortem examination with extended sampling. Infectious diseases were a major cause of death (34.7%). The number of juveniles found exceeded the adult ones. The main pathomorphological findings were hepatitis (32.8%), pneumonia (22.2%), nephritis (19.1%), liver necrosis (12.9%), and enteritis (40.7%). An unusual main finding was steatitis (20.9%) of unknown origin. Animals were mainly emaciated and showed high infection rates with *Eimeria* spp. (91.3%) and *Trichostrongylus* spp. (36.2%). *European Brown Hare Syndrome Virus* reached an epidemic status with few fatal infections (4.2%) and high seroprevalence (64.9%), whereas the prevalence of *Rabbit Haemorrhagic Disease Virus 2* was very low (0.8%) in hares in Schleswig-Holstein. Pathogens such as *Yersinia pseudotuberculosis* (5.9%)*, Pasteurella multocida* (0.8%), and *Staphylococcus aureus* (3.4%) only caused sporadic deaths. This study illustrates the wide distribution of various infectious pathogens with high mortality and even zoonotic potential. Infectious diseases need to be considered as an important influence on population dynamics in Schleswig-Holstein.

## 1. Introduction

The decline of European brown hare populations in Europe since the late 1960s has already been the topic of different studies, which focused on the role of the habitat [1,2,3], agriculture [3,4,5,6], weather and climate [7,8], reproduction [9], predation [10], juvenile mortality [11], and diseases [12,13]. In general, the current state of knowledge is that at least the long-term decline seems to be multifactorial and cannot be explained by a single cause [5,14]. In particular, the expansion of agricultural shaped landscape, which is akin to the original habitat of *Lepus* (*L.*) *europaeus* and so leads to the wide distribution of this species in the past, has been identified as the main driver of population decline due to increased intensification in recent years [5]. Although diseases have been categorized as rather secondary drivers for the long-term population decline of hares, some highly pathogenic infectious diseases have been discovered in recent decades [15]. *European Brown Hare Syndrome Virus* (EBHSV) was the first virus causing short-term population fluctuations [16] with the peracute death of adult European brown hares in particular, showing haemorrhagic organ alterations and necrotising hepatitis [17,18]. The other important lagovirus, *Rabbit Haemorrhagic Disease Virus 2* (RHDV2), first detected in 2010 in France [19] as a fatal disease in European rabbits (*Oryctolagus cuniculus*), causes symptoms quite similar to EBHSV and has also been described to infect different hare species including European hares [20,21,22]. Bacterial pathogens, for example *Yersinia* (*Y.*) *pseudotuberculosis* and *Pasteurella* (*P.*) *multocida*, as well as parasites such as coccidia, may also cause a high mortality rate in hares [13]. In the German federal state Schleswig-Holstein, the population of hares is relatively high compared to in other areas of Germany [23,24,25], but also, in this state hunting bag records and spotlight surveys show great regional differences in population densities and a slightly decreasing population trend [25,26]. In the past, some other studies in the area of Schleswig-Holstein showed that female fertility [27] and moderate regional climate [28] do not influence the population negatively, but factors such as fox density and a lack of diversity of field crops can correlate with the population decline [28,29]. Some other studies already focused on diseases in hares in this federal state and either investigated one specific pathogen or a pathogen group or conducted screenings for several diseases. Different parasites, mainly *Eimeria* spp. and *Trichostrongylus* (*T.*) *retortaeformis* [30,31,32,33], were detected. In some regions, *Toxoplasma* (*T.*) *gondii* and *Yersinia* spp. [29], *Treponema* spp. [30,34], EBHSV [30,35], and *Hepatitis E-Virus* [36] were found. All these studies only sampled hunted animals, probably causing biased pathomorphological findings as a result of the selection of more healthy individuals, as these are going to be part of the food chain or will be hunted for trophy reasons. Therefore, it is important to include deceased animals in health screening studies, although there are some limitations such as postmortem changes, difficult sample acquisition with random sampling, and often progressed decomposition. In some parts of Europe, including Germany, farmed or wild hares found dead have been included in studies before, confirming findings from studies conducted on hunted hares [37,38,39]. On the other hand, these studies also detected different incidences of pathogens such as, e.g., *Yersinia* spp. [37,38,40] compared to hunted hares. Furthermore, knowledge about regionally important pathogens such as *P. multocida* [41,42], *Francisella* (*F.*) *tularensis* [40,43], *Trichuris* spp., and *Passalurus* spp. [37], as well as *RHDV2* [21,40], was added. 

To the authors’ knowledge, the last regionwide study for different infectious diseases in deceased hares in the federal state of Schleswig-Holstein was conducted in 1990/91 [39]. In this study, hunted and deceased hares were investigated for parasites, bacteria, and viruses. Infectious diseases were found to be the cause of death in more than 50 percent of deceased animals. The major infectious pathogens found were *Eimeria* spp., gastrointestinal nematodes, *Yersinia* spp., and EBHSV with partly great differences between hunted and deceased hares. As this study was conducted almost 30 years ago, the following study on deceased or clinically abnormal hares was initiated in 2017 to approach the necessity of obtaining up-to-date knowledge of the current occurrence and influence of diseases and pathogens of hares from Schleswig-Holstein, to possibly estimate their impact on hare population development in this area.

## 2. Materials and Methods

### 2.1. Carcass Collection

To obtain some baseline data on occurring diseases and pathogens, carcasses of deceased or clinically abnormal hares found between 2017 and 2020 in the northernmost German federal state of Schleswig-Holstein were investigated. Hunters in particular were requested by public announcements in newsletters of statewide and regional hunting associations, talks, and mailing lists to send in or report these animals all year round to the Institute of Terrestrial and Aquatic Wildlife Research (ITAW) in Büsum, Germany. At the ITAW, the carcasses were either dissected directly or stored at −20 °C or +4 °C until necropsy. Pathomorphological, parasitological, bacteriological, mycological, and virological analyses were conducted.

### 2.2. Necropsy and Histopathology

Necropsies of all carcasses were performed according to a standard protocol, to be found in Appendix A in the supplements, referring to Siebert et al. [44]. First, the status of preservation was scored from 1 to 5 (1 = fresh, rigor mortis still persistent; 2 = good; 3 = moderate; 4 = progressed decomposition; 5 = mummified or skeletonized). According to this, we excluded organ samples from routine sampling for histopathology, which were already macroscopically of progressed decompensation and so could not be investigated sufficiently in further investigations. Complete animals were weighted and measured (total length and axillary girth), and hares were divided into two age classes: adult (≥1 year) and juvenile (<1 year) using lens weight [45] and the sign of Stroh [46] as criteria. The sign of Stroh is the epiphyseal cartilage of the ulna, which can only be palpated for the first 6–8 months of life in hares. The lens weight categorizes hares into adults (>1 year) and juveniles (≤1 year) by the threshold of 275 mg eye lens weight after standardized drying. It was recorded whether the carcass was frozen or just refrigerated for the interpretation of further analyses, e.g., the larval migration method does not work in previously frozen samples. The body condition was scored as good, moderate, or poor depending on fat depots in the retroperitoneum, mesentery, and pericardium combined with the diameter of the lumbar musculature. Animals with a good body condition revealed kidneys fully covered by fat and visible fat depots in the mesentery and pericardium, whereas animals with a moderate body condition demonstrated kidneys partially covered with fat without visible fat depots in the mesentery and pericardium. Hares without visible fat depots with or without atrophied lumbar musculature were allocated to a poor body condition. Sex was determined during necropsy by external sexual characteristics and gonads. The carcasses were examined for external lesions, and all organ systems were inspected macroscopically. 

If the conservation status permitted, samples of lung, heart, liver, spleen, kidney, muscle, retroperitoneal fat, brain, mesenteric lymph node, small intestine, large intestine, and adrenal glands were collected routinely and processed in paraffin wax as previously described [47]. Depending on histological findings, selected additional sections were stained with a periodic acid–Schiff (PAS) reaction, Ziehl-Neelsen’s, Gram’s, Grocott’s methenamine silver, and Turnbull’s blue stain according to standard laboratory protocols [48]. Additional tissues were routinely sampled for histopathology in 2020 (trachea, aorta, stomach, and pancreas) and in all years when macroscopic examination indicated alterations. 

### 2.3. Parasitology

Each organ system, but especially the gastrointestinal tract of every animal, was examined macroscopically for parasites. The skin and fur of all hares were screened for ectoparasites. For endoparasitic examination, every organ was assessed from the outside and sliced to search for intraparenchymal parasites. Ectoparasites and endoparasites found were first stored in water and later preserved in 70% alcohol. Species identification was performed at the Institute for Parasitology, University of Veterinary Medicine Hannover, Foundation, Hannover, Germany based on morphological characteristics via light microscopy. In some cases, the species differentiation was completed via genetic analysis using the Cyclo_Nad1F and Cyclo_trnNR primers to amplify the mitochondrial NADH dehydrogenase subunit 1 genes of cestodes [49] and the primers NC5 and NC2 to amplify the international transcribed spacer (ITS) region of nematodes [50] via conventional PCR. Furthermore, in some hares with behavioural abnormalities, reported by the hunters, conventional PCR was carried out on samples of different brain regions using the Toxo5 and Toxo8 primers to amplify a repetitive 529 bp DNA fragment of *T. gondii* [51,52].

Additionally, 4 g of faeces from the large intestine of every animal were analysed for excreted parasite stages. To detect eggs of nematodes and cestodes as well as protozoan oocysts, the combined sedimentation–flotation method using a saturated zinc sulphate solution (specific gravity: 1.3) as the flotation medium was carried out as previously described [53]. After flotation, the liquid surface was completely transferred onto a microscope slide with a wire loop and covered with a coverslip and examined microscopically at ×100 magnification. If possible, *Eimeria* species were identified morphologically according to Aoutil et al. [54] after they had been allowed to sporulate at room temperature. For the quantification of detected eggs or oocysts, a modified McMaster method [53] was used, and the results are given in eggs or oocysts per gram of faeces (epg or opg), respectively. Furthermore, from 2018 onwards, the Baermann migration method was performed to detect nematode larvae in faecal samples (minimum 1 g) of animals that had not been frozen before.

### 2.4. Microbiology

For microbiological analyses, standard swab samples (TS-Swab Pl-Amies, sterile; HEINZ HERENZ Medizinalbedarf GmbH, Hamburg, Germany) from the intestine and the lung were taken. In cases of macroscopic organ alterations, also, additional swabs or tissues were sampled. Until cultivation at the Institute of Hygiene and Infectious Diseases of Animals, Justus Liebig University Giessen, Giessen, Germany, swabs were stored at 4 °C and tissue samples at −20 °C.

Swabs and tissue samples were inoculated on 5% sheep blood agar (Merck Chemicals GmbH, Darmstadt, Germany), water-blue metachrome-yellow lactose agar (according to Gassner, Merck Chemicals GmbH), and *Yersinia*-selective (CIN-) agar (Merck Chemical GmbH) with *Yersinia*-selective supplement (Oxoid Deutschland GmbH, Wesel, Germany) immediately after arrival in the laboratory. Plates were incubated for 24 h and 48 h at 37 °C. For the detection of anaerobic bacteria, especially *Clostridium* (*Cl.*) *perfringens*, samples were streaked on Zeissler agar (Merck Chemicals GmbH) and Schaedler agar with 5% sheep blood (Becton and Dickinson, Heidelberg, Germany) and were incubated for 72 h at 37 °C under anaerobic conditions in a jar using Anaerogen^TM^ gas sachets (Oxoid).

Additionally, *Brucella* agar base with *Brucella*-selective supplement (Oxoid) and Kimmig agar (Merck Chemicals GmbH) for the selective cultivation of fungi were incubated according to the respective culture condition required for each organism. For example, *Brucella* plates were incubated in a CO_2_ incubator with 10% CO_2_ over five days. For the isolation of fungi, Kimmig agar plates were incubated at 28–30 °C for three days [55,56,57]. For bacterial enrichment, swabs were incubated in Standard I nutrient broth (Merck Chemicals GmbH) for 24 h at 37 °C and then streaked on 5% sheep blood agar (Merck Chemicals GmbH) and Gassner agar (Merck Chemicals GmbH) followed by incubation for 24 h at 37 °C. Additionally, the enrichment of sample material from the intestine was conducted by incubating it in tetrathionate broth according to Müller–Kaufmann (Merck Chemicals GmbH) and Rappaport-Vassiliadis broth (Oxoid) at 37 °C and 43 °C. After 24 h and 48 h of incubation, each broth was plated on Gassner agar (Merck Chemicals GmbH) and Brilliance *Salmonella* agar (Oxoid) and incubated at 37 °C and 43 °C for 24 h.

Grown colonies were counted to estimate bacterial growth semiquantitatively. The growth of 1–5 colonies, (+), was regarded as very isolated bacterial growth, 6–50 colonies, +, as isolated bacterial growth, 51–200 colonies, ++, as moderate bacterial growth, and >200 colonies, +++, as strong bacterial growth. Grown colonies were subcultivated and pure cultures were identified using standard morphological and biochemical methods [55,56,57] as well as MALDI-TOF mass spectrometry analysis. For this purpose, bacterial isolates were incubated on blood agar containing 5% defibrinated sheep blood (Merck Chemicals GmbH) for 24 h. Bacteria were transferred to steel targets according to the manufacturer’s instructions (Bruker Biotyper, Bruker Daltonics, Bremen, Germany) using the direct transfer protocol. Analysis was performed on a MALDI-TOF MS Biotyper Microflex LT, library version 3.3.1.0.

Hares were tested for *F. tularensis* at the Institute of Bacterial Infections and Zoonoses, Federal Research Institute for Animal Health, Jena, Germany. Liver and spleen tissue samples of each animal were analysed using cysteine heart agar (CHA, Becton Deckinson, BD Heidelberg, Germany) supplemented with 10% chocolatized sheep blood, ampicillin, and polymyxin B sulphate. Agar plates were incubated for at least 48 h at 37 °C with CO_2_-enriched atmosphere (5% CO_2_) and monitored for bacterial growth following the method of Tomaso et al. [58]. *F. tularensis* strains were identified using MALDI-TOF mass spectrometry and real-time PCR assays targeting the *tul4* gene [59]. A PCR assay with primer pair C1/C4 targeting the locus Ft-M19 enabled us to distinguish the two major subspecies, *F. tularensis* subsp. *holarctica* and *F. tularensis* subsp. *tularensis* [60]. For the molecular detection of *F. tularensis* with real-time PCR, DNA was extracted from spleen or liver samples (25 mg) with the High Pure PCR Template Preparation Kit^TM^ (Roche Diagnostics GmbH, Mannheim, Germany) according to the manufacturer’s instructions. 

### 2.5. Virology and Serology

Liver tissue samples and serum samples (S-Monovette® Serum-Gel, 7.5 ml, sterile, SARSTEDT AG & Co. KG, Nümbrecht, Germany) were screened for EBHSV and RHDV2 by the Istituto Zooprofilattico Sperimentale della Lombardia e dell’Emilia Romagna (ISZLER), Brescia, Italy. Serology was conducted with three ELISAs and two competitive enzyme-linked immunosorbent assays (cELISAs) using specific anti-EBHSV and anti-RHDV2 monoclonal antibodies for the estimation of total antibodies against EBHSV and RHDV2 following Velarde et al. [20] and one for EBHSV specific IgM conducted like described in Cooke et al. [61], with the difference that EBHSV reagents were used. Serological methods were equivalent to those described and used to detect anti-RHDV antibodies as described in WOAH Terrestrial Manual 2021 [62], but they employed specific immunological reagents towards EBHSV and RHDV2. EBHSV and RHDV2 titres ≥ 640 indicate a recent infection within a few months; titres ≥ 2560, a very recent infection a few weeks before sampling. ELISA titres tested for IgM anti-EBHSV ≥ 1280–2560 indicate a recent infection of an immunologically naïve animal. The RT value displays the ratio between titres against different lagoviruses (here, EBHSV and RHDV2) and helps to define which virus caused the detected antibodies. With an RT value > 4, the infection was certainly caused by EBHSV, and with 4, EBHSV is suspected. If the RT value is < 0.25, the infection was certainly caused by RHDV2, and with 0.25, RHDV2 is suspected. For values > 0.25 and < 4, no conclusive virus determination is possible. For some hares, from which blood samples could not be taken due to advanced decomposition, the liquid which arose from thawing frozen liver samples (called “liver juice”) was analysed as a surrogate for a proper serum to identify convalescent animals. According to personal unpublished results (Capucci et al.), the titres from liver juice is in average four times lower than real serum, so it was multiplied by four to obtain comparable values. The results of the serum were favoured over those of liver juice for animals in which both were analysed. Samples that showed an OD492 nm greater than 1.4 times that of the negative control in the cELISA at the first dilution (1/10) were considered ’suspicious’ of containing the virus and were therefore subjected to virological testing.

For virological tests, liver tissue samples were first screened with a sandwich ELISA for pan-lagovirus and in case of a positive result with a second sandwich ELISA for typing the virus as EBHSV or RHDV2 [62]. 

### 2.6. Statistical Analyses

Statistical analyses were conducted in RStudio version 4.2.2 with the ’prop.test’ package. We used two-sample tests for the equality of proportions to test the difference between two proportions and to assess whether there were statistically significant differences between adult and juvenile hares. The level of significance was set at *p* < 0.05.

## 3. Results

Between 2017 and 2020, carcasses of 118 European brown hares (*Lepus europaeus*, *L.e.*) were collected from eleven out of fifteen different administrative and urban districts within the German federal state of Schleswig-Holstein with a concentration along the western part of this federal state (Figure 1). For two hares, no data of origin were available. Carcasses were sent all year round, mainly in September (n = 27) and November (n = 30). According to hunters’ reports, the hares were either found dead (n = 84), killed by hunters through atlanto-occipital exarticulation (n = 13) or gunshot (n = 15) due to abnormalities or were killed by predators (dog: (n = 2); cat: (n = 3)). For one hare, no information was given. In 2017, a total of 53 animals was sampled, 9 in 2018, 19 in 2019, and 37 in 2020.

### 3.1. Signalment and Biometric Data

The sex was determined for each sampled animal (Table 1) and was overall distributed equally. During the first two years, a few more male hares were sampled (M/F 2017 (n = 53): 1.21; M/F 2018 (n = 9): 1.25), and in the last two years, more females were sampled (M/F 2019 (n = 19): 0.9; M/F 2020 (n = 37): 0.68). Age categorisation with lens weight was conducted for 113 hares, with a juvenile-to-adult-ratio (J/A) of 1.35 (Table 1). In the last year of the study, the age ratio was more balanced (J/A= 0.95). The average age of juvenile hares was 73 days (n = 64) with a minimum of just 1 and a maximum of 305 days. Following Suchentrunk et al. [45], an approximate date of birth was calculated for 57 hares, resulting in 68.4% of these presumably being born between June and September. For five hares, an age determination based on lens weight could not be conducted (“no data” in Table 1) because of missing or destroyed lenses. Following the sign of Stroh method, four of these hares were determined as juveniles and one hare as an adult. As this method just roughly distinguishes hares up to and over seven months of age, hereinafter we will just refer to the age categories obtained using the lens weight method.

In general, half of the examined hares showed a poor (n = 66, 55.9%) body condition, and adult hares were significantly (X^2^ (1, N = 113) = 27.71, *p* < 0.001) more often allocated to the poor body condition (n = 41, 85.4%) than juveniles (n = 22, 33.8%) (Table 2). 

### 3.2. Pathomorphological Findings

Regarding the status of preservation, carcasses were mainly allocated to category 2 (n = 42, 35.6%) or 3 (n = 60, 50.8%). Just 5.9% (n = 7) of the animals were fresh, 7.6% (n = 9) of the hares were in a state of progressed decomposition, and none were mummified. Conservation status permitted the assessment of routinely taken samples of the lung (n = 117), heart (n = 114), liver (n = 116), spleen (n = 108), kidney (n = 115), muscle (n = 116), retroperitoneal fat (n = 43), brain (n = 64), mesenteric lymph node (n = 107), small intestine (n = 110), large intestine (n = 111), and adrenal glands (n = 110) for histopathology. Additionally, the pancreas (n = 18), aorta (n = 28), stomach (n = 41), and trachea (n = 32) were sampled. In the following, only alterations are presented that were found in at least 10% of the investigated hares (Table 3). All changes are listed in Appendix A. 

#### 3.2.1. Alimentary System

During the macroscopic or histopathological examination of the intestine, most often, parasites were detected on or in the mucosa (n = 67, 60.4%). In 2017, parasites were found in 75% (n = 36) of the investigated intestines. Significantly more juvenile (n = 41, 69.4%) than adult animals (n = 22, 46.8%) were affected (X^2^ (1, N = 106) = 4.68, *p* = 0.015). The most frequently found parasites were different stages of protozoa, mainly coccidia (Figure 2). Another alteration was hepatitis (n = 38, 32.8%), mainly dominated by lymphocytes (n = 16, 42.1%) but also of mixed type (necrotising, histiocytic, purulent, granulomatous, and plasmacytic). Adult hares showed a hepatitis (n = 23, 47.9%) significantly more often (X^2^ (1, N = 109) = 6.40, *p* = 0.006) than juveniles (n = 14, 23.0%). Liver cell necrosis was diagnosed in 12.9% (n = 15) of investigated animals (Figure 3), and juveniles were affected significantly (X^2^ (1, N = 112) = 3.03, *p* = 0.041) more often (n = 12, 19.7%) than adults (n = 3, 6.3%). 

Out of 118 examined intestines, 40.7% (n = 48) showed changes indicative for enteritis. Catarrhal enteritis was most common (n = 41, 85.4%), followed by the mixed inflammatory type of lymphocytic–plasmacytic (n = 6, 12.5%), the necrotising (n = 4, 8.3%), and the granulomatous character (n = 3, 6.3%). Juveniles were more often affected (n = 31, 47.7%) than adults (n = 16, 33.3%), but this difference was not statistically significant (X^2^ (1, N = 113) = 1.79, *p* = 0.090). 

Pancreatic amyloidosis was seen in 11.1% (n = 2) of tested animals, one adult hare each in 2017 and 2020.

#### 3.2.2. Abdominal and Thoracic Cavity

The inflammation of fat tissue (steatitis) was diagnosed in 20.9% (n = 9) of the examined retroperitoneal fat tissue samples (Figure 4). Two additional positive samples, which were not included here, originated from subcutaneous tissue or epicardial fat, and a corresponding retroperitoneal fat tissue sample was not available. The main inflammatory type was pyogranulomatous to necrotising steatitis. The examined juveniles (n = 6, 27.2%) were more often diagnosed with steatitis than adult hares (n = 3, 16.7%), but this age difference was not statistically significant (X^2^ (1, N = 40) = 0.18, *p* = 0.338).

#### 3.2.3. Haematopoetic and Endocrine System

Splenic haemosiderosis was diagnosed in 14.8% (n = 16) of animals. Juveniles were less often affected (n = 6, 10.9%) than adults (n = 10, 21.3%). There was no significant difference for age categories (X^2^ (1, N = 102) = 1.35, *p* = 0.123).

#### 3.2.4. Respiratory System

Pneumonia was diagnosed in 22.2% (n = 26) of the examined samples. Adult hares were affected more often (n = 15, 31.3%) than juveniles (n = 11, 17.7%), without being significant (X^2^ (1, N = 110) = 2.04, *p* = 0.077). The main inflammatory type was purulent (n = 21, 80.8%), followed by the necrotising (n = 6, 23.1%), granulomatous (n = 4, 15.4%), histiocytic (n = 3, 11.5%), and lymphocytic types (n = 2, 7.7%). A mixed inflammatory type was present in 57.7% (n = 15) of animals with pneumonia.

The second most common alteration was tracheitis (n = 7, 21.9%, Figure 5). The trachea was only sampled routinely in 2020, and all cases of tracheitis were only from this sampling year (n = 7, 24.1%). Lymphocytic tracheitis was the most common type (n = 5, 71.4%), followed by plasmacytic (n = 4, 57.1%), purulent (n = 3, 42.9%), histiocytic (n = 2, 28.6%), and necrotising tracheitis (n = 1, 14.3%). In four of these animals, it was a mixed type (57.1%). Regarding the age distribution, adult hares showed (n = 6, 37.5%) tracheitis much more often than juveniles (n = 1, 6.3%), and this difference was statistically significant (X^2^ (1, N = 32) = 2.93, *p* = 0.044).

#### 3.2.5. Skin and Subcutis

Subcutaneous haematoma was the only alteration with a frequency above 10% for the organ system “skin” (n = 20, 16.9%). Those cases that could be explained by the history of hunters (e.g., shot by a hunter or killed by a hunting dog) were excluded. A few more adults were affected (n = 10, 20.8%) than juveniles (n = 9, 13.8%), and more males were affected (n = 12, 20.7%) than females (n = 8, 13.3%). Neither the detected differences for age (X^2^ (1, N = 113) = 0.53, *p* = 0.234) nor for sex (X^2^ (1, N = 118) = 0.67, *p* = 0.206) were statistically significant.

#### 3.2.6. Urinary and Genital System

Nephritis was an alteration in 22 hares (19.1%). The main inflammatory character was non-purulent (n = 17, 77.3%). Purulent nephritis was diagnosed in five animals (22.7%). Adult hares were significantly (X^2^ (1, N = 108) = 10.45, *p* < 0.001) more often diagnosed with nephritis (n = 17, 35.4%) compared to juveniles (n = 5, 8.3%). 

### 3.3. Parasitology

In total, 115 hares were tested for endoparasites by analysing faecal ingesta samples with the combined sedimentation–flotation or McMaster method. Both methods were conducted on 97 hares, while only the combined sedimentation–flotation method was used on 14 animals, and only the McMaster method was used on four hares. Furthermore, faecal samples of 15 hares that had not previously been frozen could be subjected to the Baermann migration method. Additionally, the skin and fur of all animals were screened for ectoparasites, and 13 hares were tested for *Toxoplasma* (*T*.) *gondii* via PCR.

In 8.7% (n = 10) of the examined faecal ingesta samples, no parasitic stages were detected. In the remaining 105 samples (91.3%), coccidia (*Eimeria* spp.) were identified. In 54 samples, enough faeces was available for in vitro sporulation, so that these samples could be determined at the species level (Table 4). Two-thirds of coccidia-positive hares (66.6%) were coinfected with another parasite species, mainly *Trichostrongylus* spp. (n = 38, 36.2%) or *Graphidium* spp. (n = 31, 29.5%) or a combination of both (n = 6, 5.7%). Interestingly, *Trichostronglyus* spp. was diagnosed in 2017 to 2019, but not in 2020. The distribution of infection in the age classes was evenly distributed for *Eimeria* spp. (adults: n = 44, 91.7%; juveniles: n = 56, 90.3%) and shifted towards the adults for *Trichostrongylus* spp. (adults: n = 17, 35.4%; juveniles: n = 18, 29.0%) and *Graphidium* spp. (adults: n = 18, 37.5%; juveniles: n = 13, 21.0%). The differences in age distribution were not significant for *Eimeria* spp. (X^2^ (1, N = 110) = 1.79, *p* = 0.5) and *Trichostrongylus* spp. (X^2^ (1, N = 110) = 0.26, *p* = 0.306), but significantly more adults were infected with *Graphidium* spp. (X^2^ (1, N = 110) = 2.88, *p* = 0.045).

Further detected parasite stages were *Trichuris* spp. (n = 8, 7.0%), *Passalurus* spp. (n = 3, 2.6%), *Cittotaenia* spp. (n = 1, 0.9%), and *Mosgovoyia pectinata* (n = 1, 0.9%). Lungworms were not detected in any of the samples examined with the Baermann migration method.

For 88 individuals, coccidia were detected with the McMaster method. Of those, 21 animals (23.9%) showed a relevant excretion of coccidia oocysts (Table 5), and so did 18.3% of all parasitologically tested hares. 

In 43 (41.0%) *Eimeria* spp.-positive animals, enteritis (catarrhal, necrotising, granulomatous, or lymphocytic–plasmacytic) was diagnosed. Of the hares showing high coccidian oocyst excretion (>100,000 opg), 57.1% (n = 12) had enteritis. None of the *E. stiedai*-positive animals showed typical bile duct alterations.

Detected ectoparasites were Ixodes spp. (n = 6, 5.1%) and *Haemodipsus* spp. (n = 1, 0.8%). 

All 13 samples tested for T. gondii via PCR were negative.

### 3.4. Microbiology

Different microbiological samples were taken from 113 hares, detecting 107 different species or genera of bacteria and fungi in various tissues (Appendix A). Hereinafter, only bacteria and fungi are presented (Table 6) and discussed, which were already described previously as clinically relevant pathogens in hares. *Escherichia* (*E*.) *coli* was cultured by far the most (n = 235) and in various organs, followed by *Yersinia* (*Y*.) *pseudotuberculosis* (n = 33). Other bacteria with zoonotic potential were *Pasteurella* (*P*.) *multocida* (n = 14), *Staphylococcus* (*St*.) *aureus* (n = 28), *Yersinia* (*Y*.) *enterocolitica* (n = 3), *Clostridium* (*Cl*.) *perfringens* (n = 27), *Listeria monocytogenes* (n = 2), *Pseudomonas aeruginosa* (n = 3), and *Klebsiella pneumoniae* (n = 1). Typical necrotic, granulomatous, or purulent inflammatory alterations were associated with *Y. pseudotuberculosis* infections in 19 tissue samples. Overall, 12 purulent organ alterations were caused by an infection with *St. aureus* and one purulent tracheitis with proof of *P. multocida*. In 32 cases of intestinal proof of *E. coli*, catarrhal, lymphocytic, plasmacytic, or granulomatous enteritis was diagnosed. *Cl. perfringens* was detected parallel to *E. coli* in four of these cases and in one extra animal with enteritis. In hares with catarrhal–purulent pneumonia, among others, *Fusobacterium* spp. (n = 1) or *Mannheimia granulomatis* (n = 2) was isolated from the lung or trachea.

In ten animals with various types of hepatitis, we found *Y. pseudotuberculosis* in at least one organ (five detections in the liver), and in six hares with hepatitis, *St. aureus* was isolated in at least one tissue (two detections in the liver).

Neither in culture nor with molecular analyses was *Francisella (F.) tularensis* detected in any of the 114 tested samples.

### 3.5. Virology and Serology

Out of 118 hares, 62 were tested for a current or overcome lagovirus infection in either virology, serology, or both. 

Altogether, 37 animals were tested for serology, with 15 serum and 22 liver juice samples. The overall seroprevalence for *European Brown Hare Syndrome Virus* (EBHSV) was 51.4% (n = 19), including those which had an RT value of 4, but the true one ranged up to 64.9% (n = 24). None of the tested hares had a serotitre indicating a previous *Rabbit Haemorrhagic Disease Virus* 2 (RHDV2) infection (RT < 0.25). Of those hares tested, four hares were negative for lagovirus in serology (10.8%), and for five hares, serological results were inconclusive for exact virus determination (13.5%). For 15 hares, the serology was conclusive for an overcome EBHSV infection (40.5%), including three being infected within a few months and two within a few weeks before sampling. For four hares, an overcome EBHSV infection was suspected (10.8%). 

For ten hares which were suspected in serology to have a virus, further sandwich ELISAs were performed for virology, detecting nine hares to be currently infected with EBHSV and one hare currently infected with RHDV2 at the time of death. The hare with the current RHDV2 infection was previously infected with EBHSV, as it was also positive for EBHSV antibodies in serology. In the pathomorphological investigation, this hare was diagnosed with severe hepatocellular necrosis and haemorrhages in tracheal mucosa and endocardium, suggesting a fatal RHDV2 infection.

Typical pathomorphological alterations such as hepatocellular necrosis were diagnosed in eight of these currently infected hares (RHDV2 infection: n = 1; EBHSV infection: n = 7). Seven additional animals with pathognomonic hepatocellular necrosis were either not tested for lagovirus infection (n = 4), were previously infected with EBHSV (n = 2), or had an RT < 4, where conclusive virus determination was not possible (n = 1). Furthermore, hepatitis was correlated with a current EBHSV infection (n = 1), certain EBHSV infections (n = 9), and uncertain EBHSV infections (n = 2). Following previous studies on EBHSV [12,63] a suspected chronic or subclinical form with mild lymphocytic hepatitis was detected in 26.3% (n = 5) of EBHSV-seropositive cases (RT ≥ 4).

Out of all hares virologically tested for EBHSV (n = 62), proportionally, slightly less juvenile animals (n = 15, 44.1%) had contact with EBHSV than the adult ones (n = 13, 48.1%), but this difference was not statistically significant (X^2^ (1, N = 61) < 0.01, *p* = 0.478). More juveniles were currently infected with lagovirus (n = 8, 23.5%) than adults (n = 2, 7.4%). This age difference was not significant (X^2^ (1, N = 61) = 0.09, *p* = 0.090).

### 3.6. Causes of Death

It was possible to determine a probable cause of death for 79.7% (n = 94) of hares. This included conclusive and suspected causes of death. Death was determined as conclusive in 39.0% (n = 46) of cases, if either corresponding information from the anamnesis was given (killed by a hunter or observed to be killed by a predator) or if characteristic morphological changes correlated to a particular pathogen detected in the affected organs/tissues, indicating a causal relationship. In cases with characteristic, fatal morphological tissue changes and the missing detection of corresponding pathogens or the detection of pathogens in several organs without a corresponding pathomorphology, the cause of death was suspected. Additionally, typical traumatic alterations without corresponding information in anamnesis were categorized as suspected.

Considering the findings of all analyses for each animal, the most probable and most common cause of death, including conclusive and suspected ones, was organ failure associated with fatal infection (n = 41, 34.7%). For 9.3% (n = 11), we defined a certain pathogen as the reason for conclusive fatal infection: EBHSV (n = 5, 4.2%), *Y. pseudotuberculosis* (n = 4, 3.4%), *St. aureus* (n = 1, 0.8%), and RHDV2 (n = 1, 0.8%). Suspected causes of fatal infection were endoparasites (n = 14, 11.9%), EBHSV (n = 7, 5.9%), *St. aureus* (n = 3, 2.5%), *Y. pseudotuberculosis* (n = 3, 2.5%), not determined lagovirus (n = 2, 1.7%), and *P. multocida* (n = 1, 0.8%). Proportionally, juveniles died due to fatal infections (n = 22, 33.8%) as often as adults (n = 18, 37.5%), and there was no significant age differences found (X^2^ (1, N = 113) = 0.04, *p* = 0.580).

Thirty-one animals were killed by hunters or their dogs (n = 31, 26.3%), but at least ten of those killed hares also showed evidence of infectious pathogens (endoparasites (n = 4), *St. aureus* (n = 2), *Y. pseudotuberculosis* (n = 3), EBHSV (n = 1)). Trauma with peracute bone fractures, haemorrhages, or tissue lacerations was the suspected cause of death in 15.3% (n = 18) of hares, and here, at least six animals showed evidence of infectious pathogens (endoparasites (n = 4), EBHSV (n = 2)). Four hares were probably killed by predators, and in one of these, *P*. *multocida* was found in several organs.

## 4. Discussion

The aim of this study was to investigate the current occurrence of infectious diseases in hares in the German federal state of Schleswig-Holstein and to establish their current impact on the population development of European brown hares in this area. Infectious pathogens [12,13] and environmental influences [1,2,3,4,5,6,7,8], solely or synergistically, are reasons for the decline of the hare population throughout Europe. In combination with adverse environmental conditions, even facultative pathogens may have an influence on individual health and so on regional populations [14,30]. To the knowledge of the authors, this is the first regionwide study in the federal state of Schleswig-Holstein evaluating various infectious diseases in European brown hares using deceased animals since Kwapil [39]. 

For this pilot study, the sample acquisition through collaboration with local hunters turned out to be suitable and led to a sample size of 118 deceased hares from almost all areas of Schleswig-Holstein with every study year represented. Furthermore, the used methods could clarify the cause of death in many cases. Still, it must be mentioned that at least the uneven distribution of specimen investigation over the study years was partly influenced by the study design, and so, in 2018, due to a lack of funding, just a few hares were investigated. The spatial cluster of submitted carcasses in the western part of Schleswig-Holstein was influenced by the sampling effort and needs to be interpreted with caution, as more animals from this region were collected. Local landscape structure and agricultural use, and so, the population density of hares, are quite different in this federal state [25,26,64] and may represent influencing factors, as already discussed elsewhere [28]. Higher population densities usually make the spread of infectious diseases more likely [65], which might increase their impact in more densely populated western North Sea marshes of Schleswig-Holstein, possibly leading to a larger number of submitted carcasses. Most hares were found in September and November, which, at least for November, matches with other studies [37,38,39]. This may be again due to a sampling bias, because at this time of the year, fields are harvested, raising the likelihood of finding dead hares. Furthermore, hunters are probably more frequent in the hunting grounds during the main hunting season for *L. europaeus* (01.10–31.12). Alternatively, harsh weather conditions with rain and cold might weaken the immune defence and so promote infectious diseases. In this main sampling period of the year, less frequently occurring diseases such as pseudotuberculosis and pasteurellosis are probably underrepresented, because these are more often reported during winter and spring [13,66]. 

In our study, the majority of deceased hares were juveniles, with an almost consistent age distribution over all study years. According to Zörner [67], in normal hare populations, juveniles exceed adults. Other studies on deceased hares [37,38,39,40], but also those conducted on hunting bags [30,68], exhibit an inverse age distribution to this, with more adults than juveniles, as determined in our study for 2020. Kugel [38] also investigated hunted animals and found hunting grounds with above-average population densities to have an increased juvenile ratio towards below-average ones. On the other hand, it is known that there is a very high juvenile mortality rate within hares, with up to 95% of them not surviving their first year [69,70]. As our study investigated deceased animals, the higher proportion of juveniles supports this reported high juvenile mortality.

More than half of the animals were malnourished with 85.4% of total adults and 73.0% of all examined animals from 2020. As in 2020, proportionally more adults were sampled than in the other years; this could also be an effect of age. Poor body condition could be caused by insufficient food supply [71] or by subacute to chronic diseases. In our study, almost all hares were diagnosed with the excretion of parasites, especially *Eimeria* spp. (91.3%), *T. retortaeformis* (36.2%), and *Graphidium* spp. (29.5%), which can cause emaciation [72]. Poor body condition score in a high proportion of deceased or euthanized hares was confirmed by another study from the Netherlands [42]. This Dutch study also showed a high infection frequency with endoparasites (e.g., *Eimeria* spp. = 66.7%), but parasites of the alimentary and respiratory tract were considered as the cause of death in only 18.0% of hares. In our study, a high percentage of analysed hares was diagnosed with relevant coccidia excretions, and 57.1% of these cases were diagnosed with enteritis in the pathomorphological investigation. These correlating enteritides were mainly of the catarrhal type, which causes malabsorption and so could be a possible reason for emaciation, as previously described [15,72]. Otherwise, more than 50% of coccidia-positive animals did not have changes indicative for enteritis. So, it remains unclear whether this is rather a normal, tolerated parasite load and just becomes relevant for individual health in combination with other infectious diseases for immunocompromised individuals [13] or under harsh weather conditions [73], or whether it has a general influence on the population. Although we found *E*. *stiedai* in almost 30% of identifiable samples, no correlating morphological alterations were observed in the liver. Concerning our findings with high infection rates of *T*. *retortaeformis*, another study from Scotland conducted on mountain hares (*Lepus timidus*) detected that an induced reduction in this nematode led to the increased fecundity of female hares [74]. To our knowledge, only regional studies on the islands Pellworm and Föhr in Schleswig-Holstein looked deeper into the fecundity of hares [27,30] without any evidence of reduced reproductivity. However, previous studies also confirm our results of high infection rates of *T. retortaeformis* in this federal state [31,32], and so, further research will be necessary to elucidate a probable causality with reproduction in this area.

Organ failure caused by infectious diseases was the suspected cause of death in one-third of investigated hares in our study. This would be even more if including those which were diseased severely but killed by other reasons. Kwapil [39] found a higher total proportion of fatal infectious diseases in roughly half of investigated hares, and contrary to our study, proportionally more juveniles died due to infectious diseases compared to adults. The infectious pathogens found were mainly the same in our study. We did not find Salmonella spp., and our method was not convenient to detect trematodes, but instead, we detected a lethal RHDV2 infection in one hare in Schleswig-Holstein. Since the first described infections of hares in Italy [75] and Spain [20], cases in France [21] and Australia [22] were reported, especially where hares live in close contact to rabbits. In wild rabbits (*Oryctolagus cuniculus*) in Germany, the nationwide dissemination of RHDV2 has already been known for many years, with several cases also being detected in Schleswig-Holstein. For a few years, RHDV2 has also been described in hares from Germany [76], but not yet in Schleswig-Holstein. To identify hotspots for possible interspecific transmission events of RHDV2 between rabbits and hares, we sampled a few deceased or euthanized rabbits parallel to this study for virology and pathology and found five out of eight to be infected by RHDV2. Spatiotemporally, no direct transmission from rabbits to hares could be concluded, because the positive cases of rabbits and the positive hares were locally separated. Our virological investigations indicate that the prevalence of RHDV2 among hares is very low, and this virus is not actively circulating in the hare population in Schleswig-Holstein. Based on these results, we conclude that there is no transmission between hares in this federal state, and those rare cases, as detected in our study, are due to transmission from rabbits in situations of high infection pressure and reduced immune defence. This is comparable to the situation described in Italy and Spain [20], where only very few wild rabbit populations exist (continental Italy) or the distribution of brown hares is limited (Catalonia, Spain). Contrary to this, in France, the mortality of hares infected with RHDV2 was found to be quite high [21], as a likely consequence of high-density populations of rabbits and brown hares living in sympatry. The other important lagovirus for hares, EBHSV, caused death less often in hares in our study compared to Kwapil [39]. This was expected considering that at the time the study of Kwapil was conducted, EBHSV first entered the country [77], finding a population completely naïve and receptive to the infection. Nowadays, the seroprevalence is high (51.4% to 64.9%), indicating a stable circulation of the virus in the populations. Previous studies in this federal state with data from 1998–2000, 1989/90, and 2010/11 describe average seroprevalences of 29% [29], 58.8% [35], and even 78.1% [30], respectively, with immense differences between years and with a significant positive correlation to population density. A low level of incidence of the deadly disease and a high level of seroprevalence are indicative of an endemic status for EBHSV and likely reflect a good density of hare populations, at least according to the model of the diffusion of EBHSV suggested by Salvioli et al. [78]. Through simulations, they showed that EBHSV transmission has complex dynamics, strongly affected by hare density. In particular, a density threshold of seven individuals/km^2^ was identified, determining two opposite epidemiological patterns: the extinction of EBHSV below the threshold and its endemic stability when the hare population density is above the threshold, with a seroprevalence proportional to the population density. As a consequence of the absence of viral circulation, all of the animals rapidly become susceptible to the infection, and the population is likely to be exposed to recurrent EBHS outbreaks due to reintroductions of the virus. Therefore, as claimed already in Italy [79], EBHS might also become more important for the hare population in Schleswig-Holstein in the next few years in the case of a drastic reduction in seroprevalence as a consequence of an undulating and decreasing population trend in Schleswig-Holstein [25].

Yersiniosis is another disease that was previously linked to high mortality rates in hares and was diagnosed with density-dependent seroprevalences of up to 55% in Schleswig-Holstein in former studies [29]. Kwapil [39] correlated 11.25% of total mortality events to *Y*. *pseudotuberculosis*, but in our study, up to 5.9% of investigated hares were diagnosed with lethal yersiniosis. This indicates that yersiniosis as an ubiquitous environmental organism [13] currently seems to be of less importance for hares in this federal state. To determine whether antibodies against *Yersinia* spp. are associated with those against EBHSV as described previously [29], serological investigations for *Yersinia* spp. should be conducted in future studies. 

*Staphylococcus aureus* and *Pasteurella multocida* caused only sporadic deaths, whereas epidemic population mortalities of up to 80% were described elsewhere [13]. 

For some morphological alterations, no correlation was found to one of the investigated pathogens. So, for example, the aetiology of the relatively often occurring granulomatous-necrotising steatitis found in 20.9% of examined samples remains unresolved. In this study, due to a standardized sampling protocol, only retroperitoneal fat was sampled and was often due to severe emaciation only present in a few animals. Nevertheless, unpublished data from a pathomorphological investigation of better-nurtured, hunted hares from the same federal state suggest that steatitis is not just limited to retroperitoneal fat tissue but also occurs in other fat depots in the body, indicating pansteatitis instead, and therefore a more generalized cause. Similar alterations have already been described in various species of reptiles [80,81], birds [82], fish [83,84], and mammals [85,86,87,88], even including rabbits [89] and European brown hares [90]. In these previous studies, several reasons such as infections, vasculitis, neoplasia, trauma, nutritional deficiencies of selenium and vitamin E, toxicosis, environmental factors, or immune-mediated diseases were proposed as possible causes. In rabbits and hares, lesions called yellow fat disease were described as an accumulation of lipogenic pigment in fat cells [90], and to be reproducible, at least in rabbits, the rabbits must be fed high amounts of unsaturated fatty acids in the absence of vitamin E, and they can also be prevented through vitamin E supplementation [91]. Still, our results which mainly display a granulomatous-necrotising type of steatitis are slightly different from these findings and need to be elucidated in detail in future studies.

Other common morphological alterations which remain at least partly unresolved were inflammations of kidneys, lungs, and livers. For some of these, the aetiology may be explained by bacterial or viral infections of the respective organs. However, for others, no possible causative pathogen was found. Cases of pneumonia were purulent in 80.8%, and so a bacterial infection, although not always detected in this study, was likely. For inflammatory liver alterations, an involvement of lymphocytes was most frequent (42.1%), and adults were affected significantly more often. Lymphocytic hepatitis was correlated in former studies with a suspected chronic form of EBHSV infections [12,63]. In our study, we detected this inflammatory type of liver together with a positive serotitre for EBHSV in 26.3% of all seropositive animals. So, this common type of hepatitis can indicate an even higher seroprevalence for EBHSV, which probably could not be confirmed by virology due to methodical inaccuracy in our study, for example, due to the analysis of liver juice instead of serum. Cases of nephritis occurred significantly more often in adults, suggesting a rather chronic impact. The mainly lymphocytic character (63.6%) supports this interpretation. Similar alterations of kidneys were detected in infections with *Encephalitozoon* spp. [42,92], which we did not test for in this study. As these species are zoonotic agents [93], the occurrence of nephritis in European brown hares needs to be investigated for this pathogen in future studies.

This study gives an up-to-date insight into the current occurrence and influence of pathogens and infectious diseases of deceased hares in the German federal state of Schleswig-Holstein. We demonstrate that various highly infectious and lethal pathogens and pathologies are present in hares in Schleswig-Holstein and so could have the potential to influence local European brown hare populations. Investigations on deceased animals are valuable to gain basic knowledge, but further studies should include toxicology and serological investigations of hunted hares to complement our gained data to the current health status of *L. europaeus* in Schleswig-Holstein.

## Figures and Tables

**Figure 1 pathogens-12-00239-f001:**
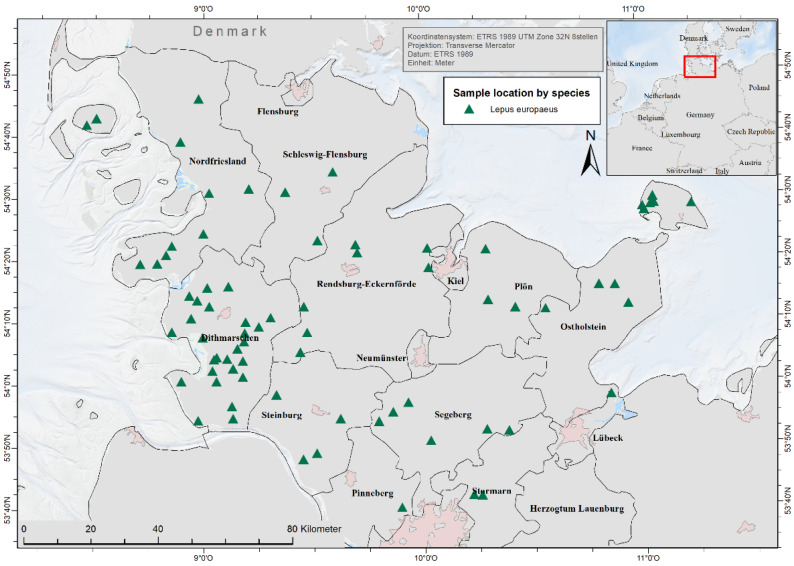
Carcass location of 116 European brown hares from 2017 to 2020 in Schleswig-Holstein, Germany.

**Figure 2 pathogens-12-00239-f002:**
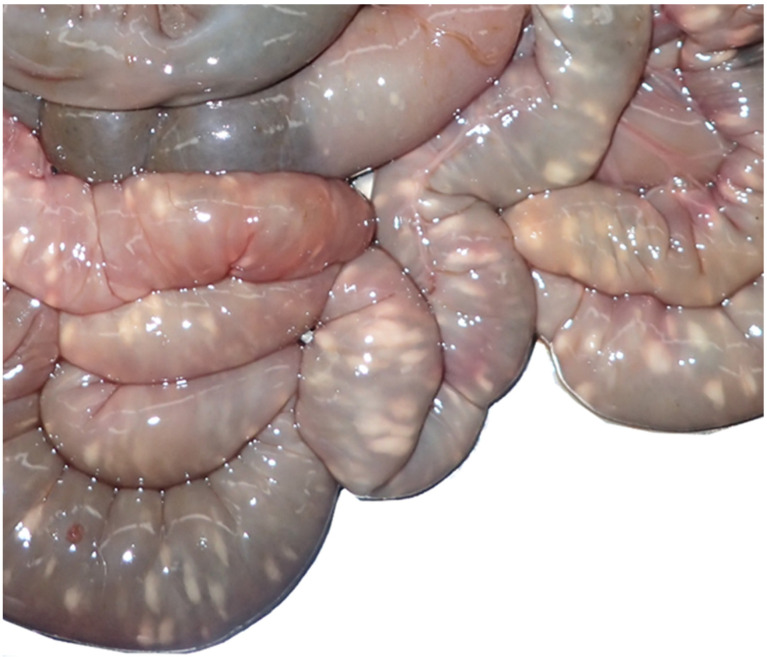
Severe coccidiosis in the jejunum of a juvenile hare (white nodules in the intestinal wall).

**Figure 3 pathogens-12-00239-f003:**
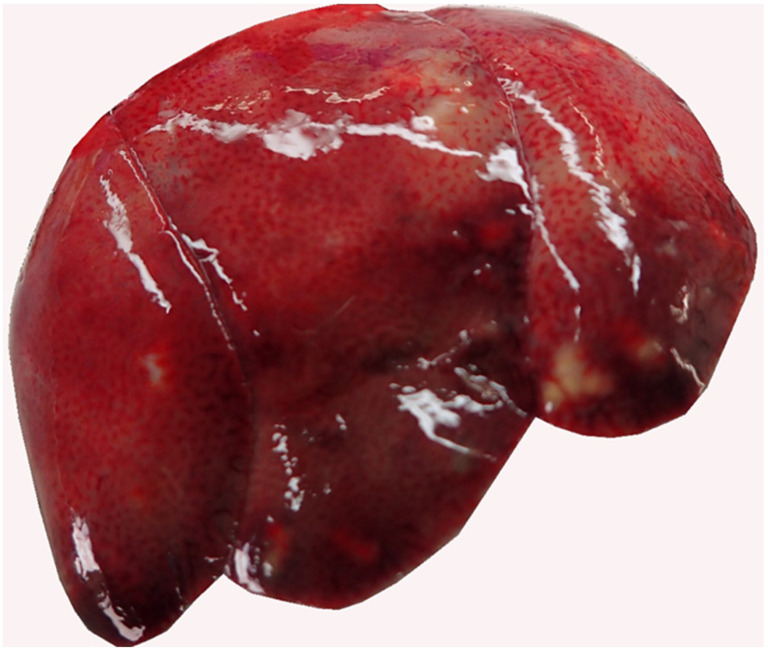
Brick-red-coloured liver in RHDV2-positive, young rabbit with hepatic coagulation necrosis (with yellow-greyish oligofocal proliferative cholangitis and fibrosing pericholangitis), representative of pathomorphological alterations seen in lagovirus-infected hares in this study.

**Figure 4 pathogens-12-00239-f004:**
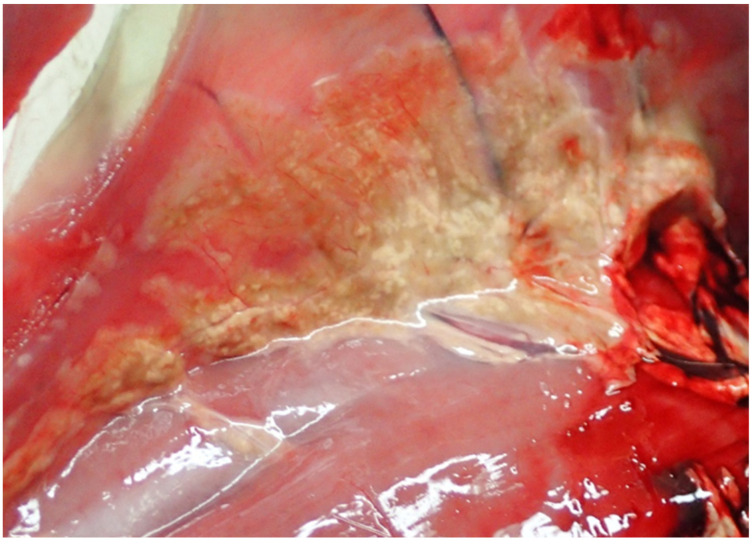
Granulomatous to necrotising steatitis of retroperitoneal fat tissue from a juvenile hare.

**Figure 5 pathogens-12-00239-f005:**
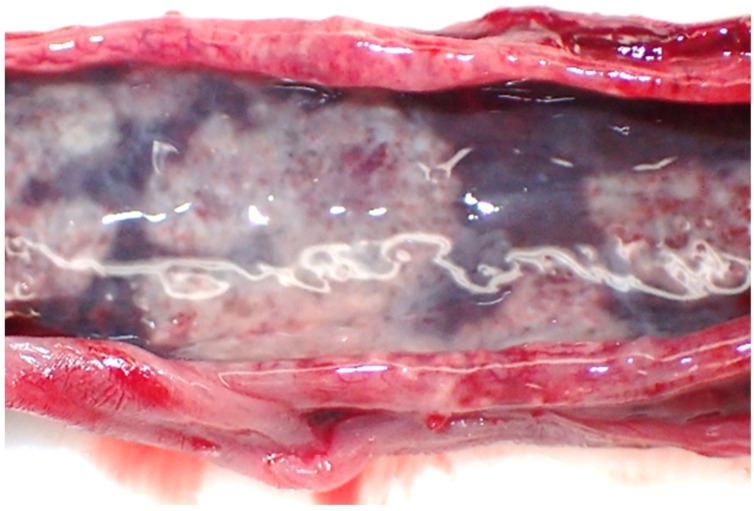
Severe catarrhal purulent tracheitis of an adult hare with systemic yersiniosis.

**Table 1 pathogens-12-00239-t001:** Sex and age distribution of sampled hares for sampling years aged with lens weight method. J: juvenile; A: adult; M: male; F: female.

	2017	2018	2019	2020	Overall
** *Lepus europaeus* **	53	9	19	37	118
adult female male	18 9 9	3 3 0	8 3 5	19 10 9	48 25 23
juvenile female male	31 14 17	5 1 4	11 7 4	18 12 6	65 34 31
no data ^1^ female male	4 1 3	1 0 1	0 0 0	0 0 0	5 1 4
J/A	1.72	1.67	1.38	0.95	1.35
M/F	1.21	1.25	0.90	0.68	0.97

^1^ no data = could not be analysed for lens weight.

**Table 2 pathogens-12-00239-t002:** Body condition of hares over sampling years by sex (n = 118) and age determined with lens weight method (n = 113).

	2017	2018	2019	2020	Overall
**good** female male juvenile adult	26 10 16 25 0	2 0 2 2 0	10 6 4 8 2	6 3 3 5 1	44 19 25 40 3
**moderate** female male juvenile adult	1 1 0 1 0	3 1 2 1 1	0 0 0 0 0	4 4 0 1 3	8 6 2 3 4
**poor** female male juvenile adult	26 13 13 5 18	4 3 1 2 2	9 4 5 3 6	27 15 12 12 15	66 35 31 22 41

The average body weight of adult hares was 3437 g (n = 48), and for juveniles, 2090 g (n = 64).

**Table 3 pathogens-12-00239-t003:** Incidence of pathomorphological findings and the yearly distribution found in at least 10% of hares.

Morphological Findings	2017	2018	2019	2020	Total	Total %
**Alimentary System**						
Enteritis	23	3	5	17	48	40.7
Parasites in intestine	36	2	10	19	67	60.4
Hepatitis	13	3	7	15	38	32.8
Liver necrosis	2	2	8	3	15	12.9
Pancreatic amyloidosis	1			1	2	11.1
**Abdominal and Thoracic Cavity**						
Steatitis	2	3	2	2	9	20.9
**Haematopoetic and Endocrine System**						
Splenic haemosiderosis	10	1	2	3	16	14.8
**Respiratory System**						
Pneumonia	10	2	3	11	26	22.2
Tracheitis				7	7	21.9
**Skin and Subcutis**						
Subcutaneous haematoma	10	2	3	5	20	16.9
**Urinary and Genital System**						
Nephritis	8	2	5	7	22	19.1

**Table 4 pathogens-12-00239-t004:** Distribution of coccidia species (*Eimeria* (*E*.) spp.).

Coccidia Species (*Eimeria* spp.)	Total	Total %
*E. leporis leporis*	20	37.0%
*E. stiedai*	16	29.6%
*E. gantieri*	14	25.9%
*E. leporis brevis*	13	24.1%
*E. deharoi deharoi*	7	13.0%
*E. cabareti*	6	11.1%
*E. tailliezi*	6	11.1%
*E. nicolegerae*	6	11.1%
*E. audubonii*	5	9.3%
*E. orbiculata*	5	9.3%
*E. reniai*	3	5.6%
*E. macrosculpta*	2	3.7%
*E. bainae*	2	3.7%
*E. deharoi rotonda*	2	3.7%
*E. europaea*	2	3.7%
*E. mazierae*	1	1.9%

**Table 5 pathogens-12-00239-t005:** Distribution of relevant coccidia excretion (>100,000 opg) detected in the McMaster method by age, sex, and year among all tested animals (n = 101).

	2017	2018	2019	2020	Total Detected
**Juvenile**	31.8% (7/22)	0.0% (0/4)	18.2% (2/11)	41.2% (7/17)	29.6% (16/54)
**Adult**	23.1% (3/13)	0.0% (0/3)	0.0% (0/8)	5.3% (1/19)	9.3% (4/43)
**Male**	31.8% (7/22)	0.0% (0/4)	22.2% (2/9)	26.7% (4/15)	26.0% (13/50)
**Female**	25.0% (4/16)	0.0% (0/4)	0.0% (0/10)	19.0% (4/21)	15.7% (8/51)
**Total**	28.9% (11/38)	0.0% (0/8)	10.5% (2/19)	22.2% (8/36)	

**Table 6 pathogens-12-00239-t006:** Relevant bacterial and fungal microorganisms with regard to their organ localization in deceased hares.

Bacteria/Fungi	Brain	Heart	Intestine	Kidney	Liver	Lung	Mesenteric lymph node	Reproductive system	Skeletal muscle	Skin	Spleen	Trachea	Total
*Clostridium perfringens* (non-typed)	3		12	1	3	4	2				2		**27**
*Clostridium sordellii*			1										**1**
*Escherichia coli*	7	1	72	14	23	67	12	3	4		18		**221**
*Escherichia coli* var. *haemolytica*			8		1	1	1		3				**14**
*Fusobacterium* spp.						1						1	**2**
*Klebsiella pneumoniae*						1							**1**
*Klebsiella* spp.						3					1		**4**
*Listeria monocytogenes*					1	1							**2**
*Mannheimia granulomatis*				1	1	2					1		**5**
*Pasteurella multocida*	1		1		1	7	1				2	1	**14**
*Pseudomonas aeruginosa*						2			1				**3**
*Staphylococcus aureus*	1	1		4	2	8			4	6	2		**28**
*Yersinia enterocolitica*						3							**3**
*Yersinia pseudotuberculosis*			8	4	6	8	2	1		1	3		**33**
**Total**	**12**	**2**	**102**	**24**	**38**	**108**	**18**	**4**	**12**	**7**	**29**	**2**	**358**

## Data Availability

The data presented in this study are available in this published article and its supplementary information files.

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
