# Peer review of "Status of Infectious Diseases in Free-Ranging European Brown Hares (*Lepus europaeus*) Found Dead between 2017 and 2020 in Schleswig-Holstein, Germany"

_pathogens, 2023, doi:10.3390/pathogens12020239_

Round 1

Reviewer 1 Report

The main aim of the work is to investigate the current occurrence of infectious diseases in hares in the German federal state of Schleswig-Holstein. The work provides a lot of information about hare pathogens from the state of Schleswig-Holstein and it is interesting. However, it is necessary a major revision. Some of the results are descriptive and should be verified by statistical tests. Some sentences should be accompanied by references. Other sentences are contradictory.

Results

-    Figure 1. Please, add to the figure 1 a small map where the study region can be seen in a wider European framework. This would help non-European readers to better locate the area where the study was done.

-    Page 7. The information in the section 3.1. Signalment and biometric data is reiterative with table 1, it should be summarized a lot.

-    Page 9. Rather than deficient nutritional status, we should talk about body condition.

-    Page 9. It is not clear how nutritional status is determined

-    Page 9. It would be necessary to analyze whether there are significant differences.

Adult hares showed a hepatitis (n = 23, 47.9%) more often than juveniles (n = 14, 23.0%) and the incidence increased slightly over the years (2017: n = 13, 25.5%; 2018: n = 3, 33.3%; 2019: n = 7, 36.8%; 2020: n = 15, 40.5%)

-    Page 11. It would be necessary to analyze whether there are significant differences.

Juveniles were more often affected (n = 31, 47.7%) than adults (n = 16, 33.3%).

-    Page 11. The 2018 and 2019 samples are too small to draw conclusions.

In the sampling years 2020 (n = 17, 45.9%) and 2017 (n = 23, 43.4%) a higher incidence of enteritis was detected than in 2018 (n = 3, 33.3%) and 2019 (n = 5, 26.3%).

-    Page 11. The samples are too small to draw conclusions.

Examined juveniles (n = 6, 27.2%) were more often diagnosed with steatitis than adult hares (n = 3, 16.7%). Regarding sampling year distribution, 2018 had the most cases (n = 3, 50.0%), followed by 2019 (n = 2, 28.6%), 2017 (n = 2, 16.7%), and 2020 (n = 2, 11.1%).

-    Page 14. Please, write the names of the genera of the bacteria that have not been mentioned before.

-    Page 15. It would help reading comprehension if you wrote the full name of the virus before its abbreviation European Brown Hare Syndrome Virus (EBHSV).

Discussion

-    Page 17. What is this statement based on? Please, add references.

Facultative pathogens in combination with adverse environmental conditions weaken the immune system.

-    Page 17. What is the main hunting season?

Furthermore, hunters are probably more frequent in the hunting grounds during the main hunting season for L. europaeus.

-       Page 18. Please, replace malnutrition by body condition

-       Page 18. How is it that the frequency of Eimeria excretions was 91.3% and the infection frequency was 66.7%?

-    Page 19. This sentence is contradictory with the other sentences

This was expected considering that at the time this study was conducted EBHSV first entered the country [76], finding a population completely naïve and receptive to the infection.

Previous studies in this federal state describe average seroprevalences of 29% [29], 58.8% [35], and even 78.1% [30] with immense differences between years and with a significant positive correlation to population density.

A low level of incidence of the deadly disease and high level of seroprevalence are indicative of an endemic status for EBHSV, and likely reflect a good density of hare populations at least according to the model of diffusion of EBHS suggested by Salvioli et al. (2017) [77].

Author Response

Reviewer 1:

  • Results - Figure 1. Please, add to the figure 1 a small map where the study region can be seen in a wider European framework. This would help non-European readers to better locate the area where the study was done.

We added an additional small map including Germany and its bordering countries in the right upper corner of the original map for better orientation within Europe.

  • Results - Page 7. The information in the section 3.1. Signalment and biometric data is reiterative with table 1, it should be summarized a lot.

Table 1 should give an easy access to the reader about the composition of the sampled hares. In these additional 4 sentences, which your comment may refer to, we additionally highlight the most important differences. Therefore, we think it is important to leave this additional information in the manuscript.

  • Results - Page 9. Rather than deficient nutritional status, we should talk about body condition.

To keep the manuscript consistent, we replaced the wording “nutritional status” to “body condition” in the whole manuscript.

  • Results - Page 9. It is not clear how nutritional status is determined

We added in the Material and Methods section (2.2 Necropsy and Histopathology) the following sentences to specify the evaluation of nutritional status: “Animals with a good nutritional status revealed kidneys fully covered by fat and visible fat depots in mesentery and pericardium, whereas animals with a moderate nutritional status demonstrated kidneys with partly cover of fat without visible fat depots in mesentery and pericardium. Hares without visible fat depots with or without atrophied lumbar musculature were allocated to a poor nutritional status.”

  • Results - Page 9. It would be necessary to analyze whether there are significant differences.

“Adult hares showed a hepatitis (n = 23, 47.9%) more often than juveniles (n = 14, 23.0%) and the incidence increased slightly over the years (2017: n = 13, 25.5%; 2018: n = 3, 33.3%; 2019: n = 7, 36.8%; 2020: n = 15, 40.5%)”

We conducted statistical analyses for all comparisons of age categories by using 2-sample tests for equality of proportions to test the difference between 2 proportions. Differences between age categories are significant and more adults were diagnosed with hepatitis (p = 0.006). Additionally, we excluded all sampling year comparison because the differences in sample size are too high between the years for meaningful significance testing. We have taken an additional co-author, Dr. Joseph Schnitzler, who conducted the statistical analyses.

  • Results - Page 11. It would be necessary to analyze whether there are significant differences.

“Juveniles were more often affected (n = 31, 47.7%) than adults (n = 16, 33.3%).”

We conducted statistical analyses for all comparisons of age categories by using 2-sample tests for equality of proportions to test the difference between 2 proportions. Differences in age categories for enteritis were not significant (p = 0.090).

  • Results - Page 11. The 2018 and 2019 samples are too small to draw conclusions.

“In the sampling years 2020 (n = 17, 45.9%) and 2017 (n = 23, 43.4%) a higher incidence of enteritis was detected than in 2018 (n = 3, 33.3%) and 2019 (n = 5, 26.3%).”

We excluded all sampling year comparison because the differences in sample size are too high between the years.

  • Results - Page 11. The samples are too small to draw conclusions.

“Examined juveniles (n = 6, 27.2%) were more often diagnosed with steatitis than adult hares (n = 3, 16.7%). Regarding sampling year distribution, 2018 had the most cases (n = 3, 50.0%), followed by 2019 (n = 2, 28.6%), 2017 (n = 2, 16.7%), and 2020 (n = 2, 11.1%).”

We conducted statistical analyses for all comparisons of age categories by using 2-sample tests for equality of proportions to test the difference between 2 proportions. Differences between age categories are not significant (p = 0.3378). Additionally, we excluded all sampling year comparison because the differences in sample size are too high between the years for performing meaningful significance tests.

  • Results - Page 14. Please, write the names of the genera of the bacteria that have not been mentioned before.

We changed it according to your comment, although we introduced the corresponding abbreviations of the genera names already before in the text (Introduction and section 2.4). Nevertheless, for better understanding we now introduced it again in the section 3.4.

  • Results - Page 15. It would help reading comprehension if you wrote the full name of the virus before its abbreviation European Brown Hare Syndrome Virus (EBHSV).

Yes, you are right. That’s why we introduced this abbreviation in the introduction on page 2. But for better understanding we now introduced it again in the results section 3.5.

  • Discussion - Page 17. What is this statement based on? Please, add references.

“Facultative pathogens in combination with adverse environmental conditions weaken the immune system.”

We have rewritten the sentence: “Infectious pathogens [12,13] and environmental influences [1-8], solely or synergistically are reasons for the decline of the hare population throughout Europe. In combination with adverse environmental conditions even facultative pathogens may have an influence on individual health and so on regional populations [14,30].”

  • Discussion - Page 17. What is the main hunting season?

“Furthermore, hunters are probably more frequent in the hunting grounds during the main hunting season for L. europaeus.”

We included the hunting season (01.10-31.12) and changed the sentence to: “Furthermore, hunters are probably more frequent in the hunting grounds during the main hunting season for L. europaeus (01.10-31.12).

  • Discussion - Page 18. Please, replace malnutrition by body condition

We replaced “malnutrition” with “poor body condition”.

  • Discussion - Page 18. How is it that the frequency of Eimeria excretions was 91.3% and the infection frequency was 66.7%?

The 91.3% is the Eimeria excretion in our study and the 66.7% is from a reference study from the Netherlands, to which we try to compare our data, and which is introduced in the previous sentence. To make the separation clearer, we included “Dutch” in the sentence with the 66.7%:

In our study, almost all hares were diagnosed with excretion of parasites, especially Eimeria spp. (91.3%), T. retortaeformis (36.2%) and Graphidium spp. (29.5%), which can cause emaciation [71]. Poor body condition score in a high proportion of deceased or euthanized hares is confirmed by another study from the Netherlands [42]. This Dutchstudy also shows a high infection frequency with endoparasites (e.g., Eimeria spp. = 66.7%), but parasites of the alimentary and respiratory tract were considered as the cause of death in only 18.0% of hares.”

  • Discussion - Page 19. This sentence is contradictory with the other sentences

“This was expected considering that at the time this study was conducted EBHSV first entered the country [76], finding a population completely naïve and receptive to the infection.

Previous studies in this federal state describe average seroprevalences of 29% [29], 58.8% [35], and even 78.1% [30] with immense differences between years and with a significant positive correlation to population density.

A low level of incidence of the deadly disease and high level of seroprevalence are indicative of an endemic status for EBHSV, and likely reflect a good density of hare populations at least according to the model of diffusion of EBHS suggested by Salvioli et al. (2017) [77].”

We included “[…] of Kwapil […]” to clarify which study this statement is referred to:

The other important lagovirus for hares, EBHSV, is less often causing death in hares in our study compared to Kwapil (1993). This was expected considering that at the time the study of Kwapil was conducted EBHSV first entered the country [76], finding a population completely naïve and receptive to the infection. Nowadays, the seroprevalence is high (51.4% to 64.9%) indicating a stable circulation of the virus in the populations. Previous studies in this federal state describe average seroprevalences of 29% [29], 58.8% [35], and even 78.1% [30] with immense differences between years and with a significant positive correlation to population density.”

Additionally, we included the years, in which the data from the referred studies were collected, to highlight when the last studies on EBHSV seroprevalence in this area were conducted and why more recent data is needed.

We also modified the final paragraph by adding the following sentence:  “As a consequence of the absence of viral circulation, all of the animals rapidly become susceptible to the infection, and the population is likely to be exposed to recurrent EBHS outbreaks due to reintroductions of the virus

Reviewer 2 Report

Pathogens- 2163462. Status of infectious diseases in free-ranging European brown hares …

This is an interesting and well written manuscript about infectious diseases found in death brown hares in a region of Germany. The information is important to understand the epidemiology of such infectious diseases and provide information about their potential zoonotic capacity.

Some minor corrections should be considered before publication:

Materials and methods

Page 3. 

2.2. necropsy and histopathology

- Delete (Figure S1) or include it in the text. 

- In the status of preservation of bodies, detail what “adjusted” means.

-Briefly describe what is the lens weight and sign of Stroh

- Any reference for nutritional status score?

Page 4

2.3 Parasitology.

(min. 1g) means minimum 1g?

Results

Page 6. Put the number 9 instead of the word nine

Page 16.  I am not quite agreeing that a fatal infection was the cause of death. The cause of death is due to organs failure. I would suggest a more appropriate term like for example, say that the massive infection was associated with the death of brown hares. 

Discussion

Page 18. It would be better to say that infectious diseases were the suspected cause of death; it is not conclusive. 

Author Response

Reviewer 2:

  • Materials and methods - Page 3. 2.2. necropsy and histopathology

- Delete (Figure S1) or include it in the text.

We included “Figure S1” in the text and where it can be found.

- In the status of preservation of bodies, detail what “adjusted” means.

We included and so specified that even routinely collected tissue samples were excluded from sampling for histopathological investigations, if these were already in necropsy of progressed decompensation status.

-Briefly describe what is the lens weight and sign of Stroh

We included the following sentences: “The sign of Stroh is the epiphyseal cartilage of the ulna, which can be palpated for the first 6-8 month of life in hares only. The lens weight categorizes hares in adults (> 1 year) and juveniles (≤ 1 year) by the threshold of 275mg eye lens weight after standardized drying.”

- Any reference for nutritional status score?

We added the following sentences to specify the evaluation of nutritional status, because it’s the method we do it at our Institute and we did not have a reference referring to: “Animals with a good nutritional status revealed kidneys fully covered by fat and visible fat depots in mesentery and pericardium, whereas animals with a moderate nutritional status demonstrated kidneys with partly cover of fat without visible fat depots in mesentery and pericardium. Hares without visible fat depots with or without atrophied lumbar musculature were allocated to a poor nutritional status.”

  • Materials and methods - Page 4 2.3 Parasitology. (min. 1g) means minimum 1g?

Yes, this means we needed at least 1 gram of faeces to perform the Baermann migration method. I changed the abbreviation “min” to minimum to make it clearer to the reader.

  • Results - Page 6. Put the number 9 instead of the word nine

We write the numbers up to number 12 as words, everything above as numbers.

  • Results - Page 16.I am not quite agreeing that a fatal infection was the cause of death. The cause of death is due to organs failure. I would suggest a more appropriate term like for example, say that the massive infection was associated with the death of brown hares.

We changed the sentence: “Considering the findings of all analyses for each animal, the most probable and most common cause of death, including conclusive and suspected ones, was organ failure associated with fatal infection (n = 41, 34.7%).”

  • Discussion - Page 18. It would be better to say that infectious diseases were the suspected cause of death; it is not conclusive.

We deleted the word “conclusive” to make it clear, that the connection between proof of infectious agents and pathological alterations for one-third of animals is just suspected. Additionally, we inserted “Organ failure caused by infectious diseases […]” considering one of the other comments mentioned before.

Round 2

Reviewer 1 Report

The authors have done all corrections and the work is good for publication.